# The Fractal Calculus for Fractal Materials

**Fakhri Khajvand Jafari [1], Mohammad Sadegh Asgari [2] and Amir Pishkoo [1,3,*]** 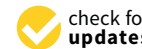

[1]  Department of Mathematics, Science and Research Branch, Islamic Azad University, Tehran, Iran; fjafari81@yahoo.com
[2]  Department of Mathematics, Central Tehran Branch, Islamic Azad University, Tehran, Iran; moh.asgari@iauctb.ac.ir
[3]  Physics and Accelerators Research School, Nuclear Science and Technology Research Institute, P.O. Box 14395-836, Tehran, Iran
*   Correspondence: apishkoo@gmail.com

**Abstract:** The major problem in the process of mixing fluids (for instance liquid-liquid mixers) is turbulence, which is the outcome of the function of the equipment (engine). Fractal mixing is an alternative method that has symmetry and is predictable. Therefore, fractal structures and fractal reactors find importance. Using $F^{\alpha}$-fractal calculus, in this paper, we derive exact $F^{\alpha}$-differential forms of an ideal gas. Depending on the dimensionality of space, we should first obtain the integral staircase function and mass function of our geometry. When gases expand inside the fractal structure because of changes from the $i + 1$ iteration to the $i$ iteration, in fact, we are faced with fluid mixing inside our fractal structure, which can be described by physical quantities $P$, $V$, and $T$. Finally, for the ideal gas equation, we calculate volume expansivity and isothermal compressibility.

**Keywords:** fractal; fractal dimension; fractal calculus; staircase function

## 1. Introduction

"Euclidean geometry" is able to model a limited number of known phenomena precisely. Introducing the concept of "fractal" provided a framework of modeling called "fractal geometry" for a large number of known phenomena in the fields of the humanities, basic sciences, medical sciences, and engineering, which have complicated forms to model [1–3]. Computer graphics has provided the possibility of making complicated and meanwhile beautiful fractal shapes by applying the mathematical language in the system of the iterative function for such shapes [4]. This prepares the ground for simulation and numerical solving of various problems with complicated geometries [3,5].

The complicated shapes of known phenomena are described with a parameter called "fractal dimension". In the engineering field, this is simply calculated by the definition of the box-counting dimension. In addition to fractal geometry, the fractional calculus and fractal calculus can also be helpful in the description of phenomena, with the difference that the operators in the latter calculi are respectively non-local and local [6–18].

Unfortunately, a fractal cannot be represented by an equation. In spite of the valuable efforts made to apply measure theory and harmonic analysis in fractals [19,20], the main step in the foundation of fractal calculus was taken by Parvate and Gangal. They suggested the algorithmic and Riemann-like method calculus on the fractal that can be mathematical models for many phenomena in nature [16,21]. By presenting an algorithmic method in fractal calculus, they introduced a proper mathematical modeling for many phenomena in nature [16,17,21]. Differentiation and integration in this method are done by introducing the step function and fractal mass function. Applying this method has led to a new formulation for Newtonian, Lagrangian, and Hamiltonian mechanics from which the Schrödinger equation for fractal curves can be solved [22].

In this paper first, the fractal dimension is computed for a specific geometric shape in the language of the iterative function system. Then, the step function in the fractal calculus is shown for the first three iterations in the algorithmic method. After that, by applying this geometry for ideal gases [23], the relations associated with exact differential equations for the physical quantities of temperature, pressure, and volume are presented.

## 2. Preliminaries

We first give the main definitions relating to the basic tools in fractal calculus.

### 2.1. The Integral Staircase Function on Cantor Cubes

Let $F$ be the triadic Cantor set. We define $\mathfrak{F} = F \times F \times F \subset \mathfrak{R}^3$ as a fractal Cantor cubes set that is the subset of $I = [a,b] \times [c,d] \times [e,f]$, $a,b,c,d,e,f \in \mathfrak{R}$ (Real-line) [24]. We plot the cross-section of the Cantor cubes with fractal dimension $\frac{\log 8}{\log 3}$ in Figure 1.

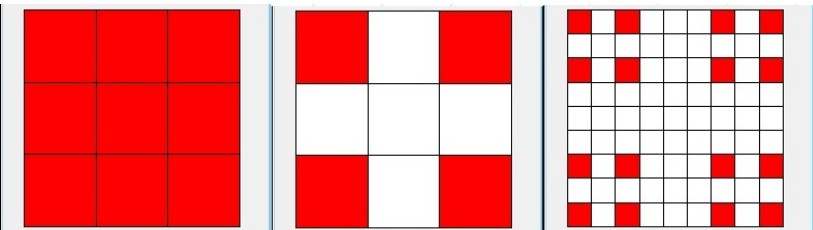

**Figure 1.** Fractal dimension of Cantor cubes = log 8/log 3 = 1.89.

The "flag function" for $\mathfrak{F}$ is defined as [18,25]:

$$\Theta(\mathfrak{F}, I) = \begin{cases} 1 & \text{if} \quad \mathfrak{F} \cap I \neq \varnothing \\ \\ 0 & \text{otherwise} \end{cases}.$$

Let us consider a subdivision of $I = [a,b] \times [c,d] \times [e,f]$ as follows:

$$P_{[a,b] \times [c,d] \times [e,f]} = \{x_0 = a, x_1, x_2, ..., x_n = b\} \times \{y_0 = c, y_1, y_2, ..., y_n = d\}$$

$$\times \{z_0 = e, z_1, z_2, ..., z_n = f\}. \tag{1}$$

The "mass function" $\gamma^{\xi}(\mathfrak{F}, a, b, c, d, e, f)$ is defined as:

$$\gamma^{\xi}(\mathfrak{F}, a, b, c, d, e, f) = \lim_{\delta \to 0} \inf_{P_{[a,b] \times [c,d] \times [e,f]} \; |P| \leq \delta} \sum_{i=1}^{n} \frac{(x_i - x_{i-1})^{\alpha}}{\Gamma(\alpha + 1)} \frac{(y_i - y_{i-1})^{\beta}}{\Gamma(\beta + 1)} \frac{(z_i - z_{i-1})^{\mu}}{\Gamma(\mu + 1)}$$

$$\times \Theta(F, [x_{i-1}, x_i]) \Theta(F, [y_{i-1}, y_i]) \Theta(F, [z_{i-1}, z_i]), \tag{2}$$

where $\xi = \alpha + \beta + \mu$ and $0 < \beta \leq 1, 0 < \beta \leq 1, 0 < \mu \leq 1$. The value of $\xi$ for the case of fractal Cantor cubes is $\xi = 0.6 + 0.6 + 0.6 = 1.8$.

Using the concept of the mass function, the "integral staircase function" for the fractal Cantor cubes $S_{\mathfrak{F}}^{\xi}(x,y,z)$ of order $\xi$ for a fractal set $\mathfrak{F}$ is defined as follows:

$$S_{\mathfrak{F}}^{\xi}(x,y,z) = \begin{cases} \gamma^{\xi}(\mathfrak{F}, a_0, c_0, e_0, x, y, z) & \text{if} \quad x \geq a_0, y \geq c_0, z \geq e_0 \\ \\ -\gamma^{\xi}(\mathfrak{F}, a_0, c_0, e_0, x, y, z) & \text{otherwise} \end{cases},$$

where $a_0, c_0, e_0$ are arbitrary real numbers. For the fractal object such as a Cantor set, we plot the staircase function for the first three iterations as they are shown in Figures 2–4, respectively.

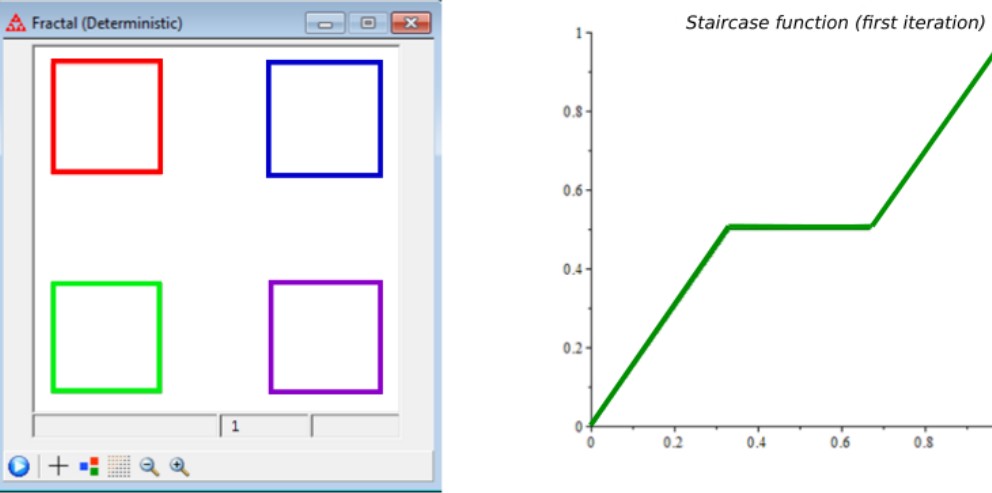

**Figure 2.** Staircase function for the first iteration.

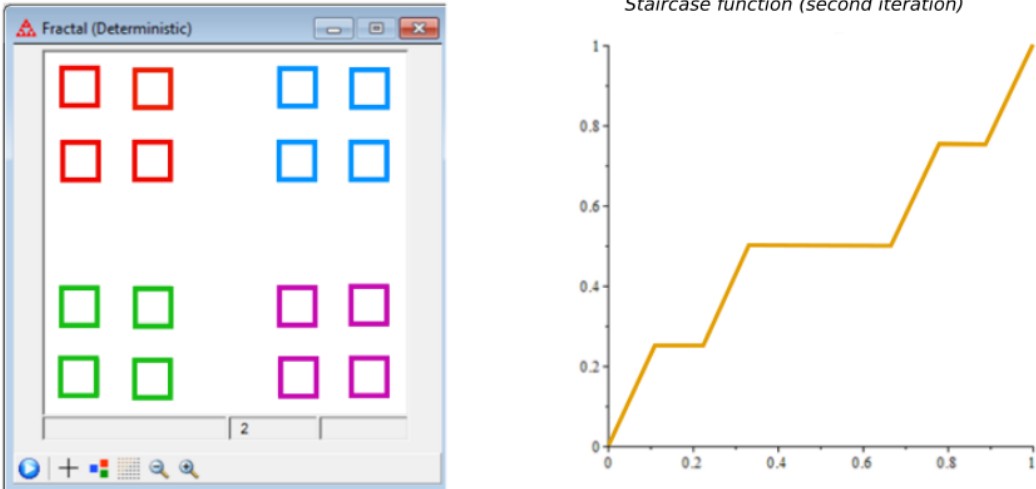

**Figure 3.** Staircase function for the second iteration.

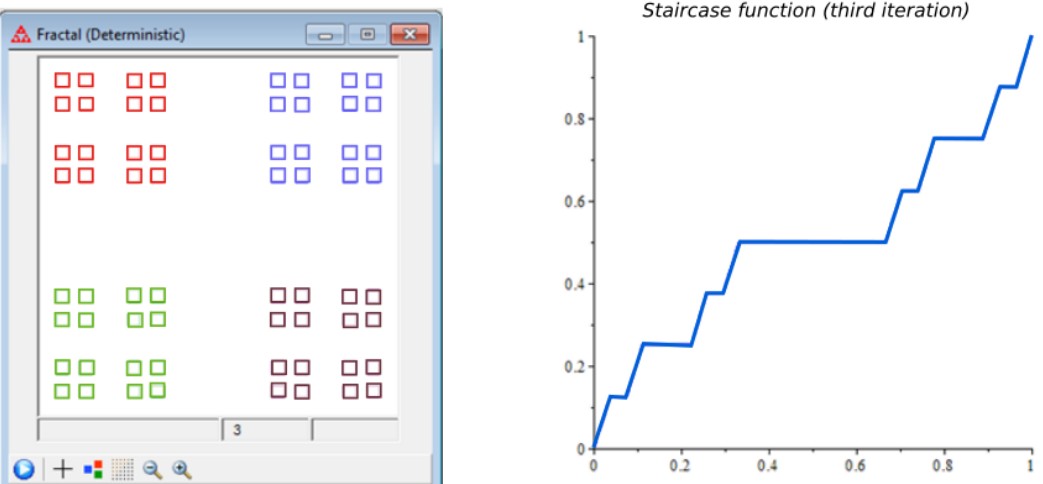

**Figure 4.** Staircase function for the third iteration.

To recall the definition of the ternary Cantor function *F* and Cantor set *C*, one can see [26].

The Cantor function $F(x)$ is defined as the function on $[0,1]$ such that for values of $x$ on the Cantor set:

**Example 1.** *Let $a_{nx} = 2$ in Equation (1.1)* [26]*. If the Cantor set is:*

$$x = \sum_{n=1}^{\infty} 2.3^{-n}$$

*then the Cantor function will be:*

$$F(x) = \sum_{n=1}^{\infty} 2^{-n}.$$

The triadic Cantor function of the fractals is a set that can be obtained by an iterative process. The first step defines a segment of unit length. The second step is to divide the segment into three equal parts of length $\frac{1}{3}$ and remove the central one. The triadic Cantor set is created by repeating this process to infinity. In general, at each stage $n$, there are $2^n$ segments with length $3^{-n}$ with $2^n - 1$ gaps between them [18].

*2.2. $F^\alpha$-Differentiation*

**Definition 1.** *[16] If F is an $\xi$-perfect set, then the $F^\xi$-partial derivative of f with respect to x is:*

$$^x D_F^\xi = \begin{cases} F\text{-}\lim_{(x',y,z) \longrightarrow (x,y,z)} \dfrac{f(x',y,z) - f(x,y,z)}{S_F^\xi(x',y,z) - S_F^\xi(x,y,z)} & \text{if } (x,y,z) \in F \\ \\ 0 & \text{otherwise} \end{cases}.$$

*If the limit exists, likewise $^y D_F^\xi f(x,y,z)$ and $^z D_F^\xi f(x,y,z)$ can be defined.*

Now, in the following, we utilized the $F^\xi$-fractional calculus on the fractal subset of $R^3$.

**Definition 2.** *[27] $F^\xi$-fractional one-forms: A differential fractional one-form on an F subset of $R^3$ is an expression $H(x,y,z)d_F^\alpha x + G(x,y,z)d_F^\beta y + N(x,y,z)d_F^\gamma z$ where $H, G, N$ are functions on the open set. If $f(x,y,z)$ is $C_\xi^1$ a function, then its $F^\xi$-fractional total differential (or exterior derivative) is:*

$$d_F^\xi f(x,y,z) = {}^x D_F^\alpha f(x,y,z)d_F^\alpha x + {}^y D_F^\beta f(x,y,z)d_F^\beta y + {}^z D_F^\gamma f(x,y,z)d_F^\gamma z,$$

*where $\xi = \alpha + \beta + \gamma$.*

**Definition 3.** *[27] $F^\xi$-fractional exactness:*
*Suppose that $Hd_F^\alpha x + Gd_F^\beta y + Nd_F^\gamma z$ is a $F^\xi$-fractional differential on F with $C_\xi^2$ function $f(x,y,z)$ with $d_F^\xi f = Hd_F^\alpha x + Gd_F^\beta y + Nd_F^\gamma z$. We will call an $F^\xi$-fractional differential closed if:*

$$^x D_F^\xi f = H \quad {}^y D_F^\xi f = G \quad {}^z D_F^\xi f = N.$$

*However, in our discussion, the main variables are $p, v, \theta$ instead of $x, y, z$, respectively. Therefore,*

$$^p D_F^\xi f = H \quad {}^v D_F^\xi f = G \quad {}^\theta D_F^\xi f = N. \tag{3}$$

*Therefore, $Hd_F^\alpha x + Gd_F^\beta y + Nd_F^\gamma z$ is exact if for variables $x, y, z$, we have:*

$$^y D_F^\beta N = {}^z D_F^\gamma G, \quad {}^x D_F^\alpha G = {}^y D_F^\beta H, \quad {}^z D_F^\gamma H = {}^x D_F^\alpha N.$$

*However, the conditions of exactness for variables pressures, volumes, and temperatures are:*

$$^{v}D_{F}^{\beta}N =^{\theta} D_{F}^{\gamma}G, \quad ^{p}D_{F}^{\alpha}G =^{v} D_{F}^{\beta}H, \quad ^{\theta}D_{F}^{\gamma}H =^{p} D_{F}^{\alpha}N. \tag{4}$$

## 3. Results

If $V$ was a geometrical quantity referring to the volume of space, then $dV$ could be used to denote a portion of that space arbitrarily small. If the change of $P$ is very small in comparison with $P$ and very large in comparison with molecular fluctuations, then it also may be represented by the differential $dP$. Every infinitesimal in thermodynamics must satisfy the requirement that it represents a change in a quantity, which is small with respect to the quantity itself and large in comparison with the effect produced by the behavior of a few molecules. We may imagine the equation of state solved for any coordinate in terms of the other two. Thus,

$$V = f_1(\theta, P).$$

A fundamental theorem in partial differential calculus enables us to write:

$$dV = \left(\frac{\partial V}{\partial \theta}\right)_p d\theta + \left(\frac{\partial V}{\partial P}\right)_\theta dP,$$

while we define its fractional counterpart:

$$d_F^{\xi} f_1 = d_F^{\xi} V = \left(^{\theta}D_F^{\xi}V\right)_P d_F^{\gamma}\theta + \left(^{P}D_F^{\xi}V\right)_\theta d_F^{\gamma}P,$$

where each partial derivative is itself a function of $\theta$ and P.

Both the above partial derivatives have an important physical meaning. If the change of temperature is made infinitesimal, then the change in volume also becomes infinitesimal, and we have what is known as the instantaneous volume expansivity, or just volume expansivity, which is denoted by $\beta$. Thus,

$$\beta = \frac{1}{V}\left(\frac{\partial V}{\partial \theta}\right)_p \quad \text{(for integer differentiation)},$$

while for the fractional case, the volume expansivity is defined as:

$$\beta = \frac{1}{V}\left(^{\theta}D_F^{\xi}V\right)_P. \tag{5}$$

As a rule, $\beta$ may be regarded as a constant within a small temperature range. The quantity $\beta$ is expressed in reciprocal degrees. The effect of a change of pressure on the volume of a hydrostatic system when the temperature is kept constant is expressed by a quantity called isothermal compressibility and is represented by the symbol $\kappa$ (Greek Kappa). Thus,

$$\kappa = -\frac{1}{V}\left(\frac{\partial V}{\partial P}\right)_\theta \quad \text{(for integer differentiation)},$$

while for the fractional case, the isothermal compressibility is defined as:

$$\kappa = -\frac{1}{V}\left(^{P}D_F^{\xi}V\right)_\theta. \tag{6}$$

Notice that if this time, the equation of state is solved for $P$, then:

$$P = f_2(\theta, V).$$

Differentiation gives:

$$dP = \left(\frac{\partial P}{\partial \theta}\right)_v d\theta + \left(\frac{\partial P}{\partial V}\right)_\theta dV,$$

and:

$$d_F^\alpha p = \left(^\theta D_F^\xi P\right)_V d_F^\gamma \theta + \left(^V D_F^\xi P\right)_\theta d_F^\beta V, \tag{7}$$

where the former is an integer differentiation and the latter is the fractional differentiation.

Finally, writing temperature $\theta$ as a function of pressure $P$ and volume $V$ gives one:

$$d\theta = \left(\frac{\partial \theta}{\partial P}\right)_v dP + \left(\frac{\partial \theta}{\partial V}\right)_p dV,$$

and:

$$d_F^\gamma \theta = \left(\frac{\partial \theta}{\partial P}\right)_v d_F^\alpha p + \left(\frac{\partial \theta}{\partial V}\right)_p d_F^\beta v. \tag{8}$$

In the most general case, if it were possible, we write:

$$d_F^\xi f(p,v,\theta) = ^p D_F^\alpha f(p,v,\theta) d_F^\alpha p + ^v D_F^\beta f(p,v,\theta) d_F^\beta v + ^\theta D_F^\gamma f(p,v,\theta) d_F^\gamma \theta, \tag{9}$$

in which $\xi = \alpha + \beta + \gamma$, and $n = 2, 1, 0$.

At each stage of the contraction of space, for gas molecules, the following formula holds:

$$N_n = 8^n,$$

$$L_n = 3^{-n},$$

$$V_n = L_n^3 N_n = \left(\frac{8}{27}\right)^n, \tag{10}$$

and vice versa for the expanding process (see Figure 5).

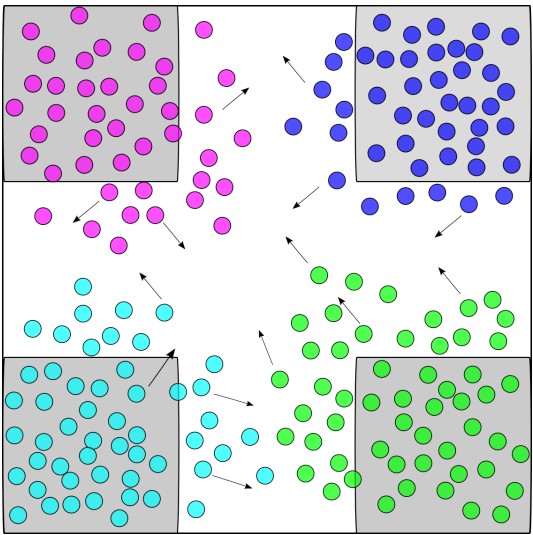

**Figure 5.** Reverse process in each iteration.

### 3.1. Equations of State

Although the three quantities temperature, pressure, and gas are sufficient to study the behavior of gases, the governing equation is simple only for the limited range, namely ideal gas. For the whole range, we should apply different equations starting from:

$$PV = nR\theta \tag{11}$$

to the Beattie–Bridgman equation:

$$P = \frac{R\theta(1 - \epsilon)}{v^2}(v + B) - \frac{A}{v^2}, \tag{12}$$

in which $A = A_0(1 - \frac{a}{v}), B = B_0(1 - \frac{b}{v}), \epsilon = \frac{c}{v\theta^3}$, and $V = \frac{v}{n}$, respectively.

Using (5) and (6), for the ideal gas equation, we calculate $\beta$ and $\kappa$ as follows, respectively:

$$\beta = \frac{1}{V}\left(^\theta D_F^\xi V\right)_P = \frac{P}{nR\theta}\frac{\partial^{0.6} V}{\partial \theta^{0.6}} = \frac{1}{\theta}\frac{\partial^{0.6}\theta}{\partial \theta^{0.6}} = \frac{1}{\theta}\frac{\Gamma(2)}{\Gamma(1.4)}\theta^{0.4} = 1.127\theta^{-0.6},$$

and:

$$\kappa = -\frac{1}{V}\left(^P D_F^\xi V\right)_\theta = -\frac{P}{nR\theta}\frac{\partial^{0.6} V}{\partial P^{0.6}} = -P\frac{\partial^{0.6} P^{-1}}{\partial P^{0.6}} = -\frac{\Gamma(0)}{\Gamma(-0.6)}P^{-0.6}.$$

## 4. Conclusions

This paper investigated fractal mixing of four different types of ideal gases inside a fractal structure by using $F^\alpha$-calculus. The flag function, mass function, and staircase function are building blocks of $F^\alpha$-calculus. Physical systems involving ideal gas are described by the equation of state, while inside the fractal object, the relations between pressure, volume, and temperature are deduced as the fractional forms here. In order to maximize symmetry and, on the other hand, minimize the unforeseeable features of mixing such as turbulence, fractal objects are a good offer.

**Author Contributions:** All authors contributed to each part of this work equally, and they read and approved the final manuscript.

**Acknowledgments:** The authors are thankful to the anonymous referees for their valuable comments and suggestions that helped to improve this paper.

**Conflicts of Interest:** The authors declare no conflict of interest.

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
