# Peer review of "The Fractal Calculus for Fractal Materials"

_fractalfract, doi:10.3390/fractalfract3010008_

Round 1

Author Response

Point 1: 1. Are you missing a comma on lines 29 and 45 at the end? 

Response 1: Yes I am sorry, the problem was resolved.

Point 2: 2. Similarly, in Definitions 1, 2 and 3, at the end of formulas (1), (2), (9) and (A2).

Response 2: Yes I am sorry, the problem was resolved.

Point 3: Please carefully typesetting the article. So much dots are missed in the end of formulas.

Response 3: Yes I am sorry, the problem was resolved.

Point 4: Page 5, line 41. It should be “If V was a geometrical quantity referring to the volume of space”.

Response 4: Yes, I corrected it.

Point 5: the manuscript badly needs an extensive editing of English language and style. As it is, it's borderline unreadable.

Response 5: Yes I have tried to solve this problem as it is highlighted.

Reviewer 2 Report

Please refer to the attached comments.

Author Response

Point 1: The abstract section is too brief and short which does not clearly indicate the purpose and importance of the research work of this paper.

Response 1: I added four sentences to clarify the subject and problem statement of our paper as it is highlighted in the paper.

Point 2: In the Introduction, there is a lack of more basic and detailed description of the fractal calculus for fractal materials. In addition, what are the problems and deficiencies in current relevant researches? The authors should make a more detailed description of the relevant literatures rather than a general discussion.

Response 2: I added more sentences to explain more about fractal calculus as it is highlighted in the paper.

Point 3: In the Preliminaries, the context is not natural and there are some spelling mistakes. Furthermore, the figures are not clear and beautiful enough.

Response 3: I raised the quality of the figures by using the software as it is highlighted in the paper.

Point 4: Some graphic images can be added to simulate and analyze the main results of this paper, so as to enrich the contributions of this paper. The current result section of this paper appears to be too single and boring.

Response 4: It is needed long time (at least 1 month) to simulate and analyze new results.

Point 5: There is a lack of necessary description and explanation of the figures and the symbols in formulas. For example, looks very strange.  .

Response 5: I clarified it as it is highlighted in the paper.

Point 6: The appendix on partial derivatives seems unnecessary.

Response 6: I deleted it as it does not exist in the paper.

Point 7: Since the main innovation of this paper lies in the application, the authors can describe the application of fractal calculus to ideal gas in more details and explain its advantages.

Response 7: I explained in the abstract as it is highlighted in the paper.

Reviewer 3 Report

The work is very interesting and has a subject of great relevance in the present day. We have the fractal calculus, on the other hand we have the fractal materials.

I really enjoyed the work. The work brings novelties, and information that holds the reader's attention. But, I have some suggestion and considerations to improvement of this work.

Some minor modifications are required, I listed below:

(1)  Put in order the references numbers in the text.

(2) I suggest the reference for the state equation (pv=nRT and the others) "

Thermodynamics and an Introduction to Thermostatistics" Callen - book, 

(3) The most sophisticate tool that derives functions in non integer order is the fractional calculus.  I will list a sequence of references on fractional calculus that will certainly help better the article, as well as attracting the attention of a different audience.

---- Approximate solutions to fractional subdiffusion equations. The European Physical Journal Special Topics, v. 193, n. 1, p. 229-243, 2011. APA

 ---- Non-Gaussian Distributions to Random Walk in the Context of Memory Kernels. Fractal and Fractional, v. 2, n. 3, p. 20, 2018. 

 ---- An introduction to the fractional calculus and fractional differential equations. 1993  (Book).

(4)  Finally, I encourage the authors to add a short conclusion, to summarize the main results.

I liked very much,

Thank you for your attention.

Author Response

Point 1: 1.  Put in order the references numbers in the text.

Response 1: I did as it is highlighted in the paper.

Point 2:  I suggest the reference for the state equation (pv=nRT and the others) "

Thermodynamics and an Introduction to Thermostatistics" Callen - book,

Response 2: I put as it is highlighted in the paper.

Point 3: The most sophisticate tool that derives functions in non integer order is the fractional calculus.  I will list a sequence of references on fractional calculus that will certainly help better the article, as well as attracting the attention of a different audience.

Response 3: I put suggested references as it is highlighted in the paper.

Point 4: Finally, I encourage the authors to add a short conclusion, to summarize the main results.

Response 4: I added the conclusion paprt in the paper as it is highlighted in the paper.

Round 2

Reviewer 2 Report

I have no other comments. The following procedures depend on the Editors.